# ENTROPY MINIMIZATION IN EMERGENT LANGUAGES

## ABSTRACT

There is a growing interest in studying the languages emerging when neural agents are jointly trained to solve tasks requiring communication through a discrete channel. We investigate here the information-theoretic complexity of such languages, focusing on the basic two-agent, one-exchange setup. We find that, under common training procedures, the emergent languages are subject to an entropy minimization pressure that has also been detected in human language, whereby the mutual information between the communicating agent's inputs and the messages is minimized, within the range afforded by the need for successful communication. This pressure is amplified as we increase communication channel discreteness. Further, we observe that stronger discrete-channel-driven entropy minimization leads to representations with increased robustness to overfitting and adversarial attacks. We conclude by discussing the implications of our findings for the study of natural and artificial communication systems.

## 1 INTRODUCTION

There has recently been much interest in the analysis of the communication systems arising when deep network agents that interact to accomplish a goal are allowed to exchange language-like discrete messages (Lazaridou et al., 2016; Havrylov & Titov, 2017; Choi et al., 2018; Lazaridou et al., 2018). Understanding the emergent protocol is important if we want to eventually develop agents capable of interacting with each other and with us through language (Mikolov et al., 2016; Chevalier-Boisvert et al., 2019). The pursuit might also provide comparative evidence about how core properties of human language itself have evolved (Kirby, 2002; Hurford, 2014; Graesser et al., 2019). While earlier studies reported ways in which deep agent protocols radically depart from human language (Kottur et al., 2017; Bouchacourt & Baroni, 2018; Chaabouni et al., 2019; Lowe et al., 2019), we show here that emergent communication shares an important property of the latter, namely a tendency towards *entropy minimization*.

Converging evidence indicates that efficiency pressures are at work in language and other biological communication systems (Ferrer i Cancho et al., 2013; Gibson et al., 2019). One particular aspect of communicative efficiency, that has been robustly observed across many semantic domains, is a tendency to minimize lexicon entropy, to the extent allowed by the counteracting need for accuracy (Zaslavsky et al., 2018; 2019). For example, while most languages distinguish grandmothers from grandfathers, very few have separate words for mother- and father-side grandmothers, as the latter distinction would make communication only slightly more accurate at the cost of an increase in lexicon complexity (Kemp & Regier, 2012). We show here, in two separate games designed to precisely measure such property, that the protocol evolved by interacting deep agents is subject to the same complexity minimization pressure.

Entropy minimization in natural language has been connected to the Information Bottleneck principle (Tishby et al., 1999). In turn, complexity reduction due to the Information Bottleneck provides a beneficial regularization effect on the learned representations (Fischer, 2019; Alemi et al., 2016; Achille & Soatto, 2018a;b). It is difficult to experimentally verify the presence of such effect in human languages, but we can look for it in our emergent language simulations. We confirm that, when relaxing channel discreteness, the entropy minimization property no longer holds, and the system becomes less robust against overfitting and adversarial noise.

## 2 GENERAL FRAMEWORK

We establish our results in the context of signaling games (Lewis, 1969), as introduced to the current language emergence literature by Lazaridou et al. (2016) and adopted in several later studies (Havrylov & Titov, 2017; Bouchacourt & Baroni, 2018; Lazaridou et al., 2018). There are two agents, Sender and Receiver, provided with individual inputs at the beginning of each episode. Sender sends a single message to Receiver, and Receiver has to perform an action based on its own input and the received message. Importantly, there is no direct supervision on the message protocol. We consider agents that are deterministic functions of their inputs (after training).

As an example, consider the task of communicating a $n$-bit number, sampled uniformly at random from $0...2^n - 1$. The full number is shown to Sender, and its $k$ ($0 \leq k \leq n$) least-significant bits are also revealed to Receiver. Receiver has to output the full number, based on the message from Sender and its own input. Would the Sender transmit the entire number through its message? In this case, the protocol would be "complex," encoding $n$ bits. Alternatively, Sender could only encode the bits that Receiver does not know, and let Receiver fill in the rest by itself. This emergent protocol would be "simple," encoding less information about the number. We find experimentally that, once the agents are successfully trained to jointly solve the task, the emergent protocol *minimizes the entropy of the messages* or, equivalently in our setup, *the mutual information between Sender's input and messages*. In other words, the agents consistently approximate the simplest successful protocol (in the current example, the one transmitting $\approx n - k$ bits).

After training, we can connect the entropies of Sender and Receiver inputs $\boldsymbol{i}_s$ and $\boldsymbol{i}_r$, messages $\boldsymbol{m} = S(\boldsymbol{i}_s)$, Receiver's output (the chosen action) $\boldsymbol{o} = R(\boldsymbol{m}, \boldsymbol{i}_r)$, and ground-truth outputs $\boldsymbol{l}$ by using standard inequalities (Cover & Thomas, 2012):

$$H(\boldsymbol{i}_s) \geq H(S(\boldsymbol{i}_s)) = H(\boldsymbol{m}) \geq H(\boldsymbol{m}|\boldsymbol{i}_r) \geq H(R(\boldsymbol{m}, \boldsymbol{i}_r)|\boldsymbol{i}_r) = H(\boldsymbol{o}|\boldsymbol{i}_r) \approx H(\boldsymbol{l}|\boldsymbol{i}_r) \quad (1)$$

(Note that, since agents are deterministic after training, $H(\boldsymbol{m}) = I(\boldsymbol{i}_s; \boldsymbol{m})$. We can then use these quantities interchangeably.) Our empirical measurements indicate that the entropy of the messages $\boldsymbol{m}$ in the emergent protocol tends to approach the lower bound: $H(\boldsymbol{m}) \to H(\boldsymbol{l}|\boldsymbol{i}_r)$, even if the upper-bound $H(\boldsymbol{i}_s)$ is far.

In our experiments, we observe that when the amount of information that Receiver needs is reduced, without changing other parameters, the emergent protocol becomes simpler (lower entropy). In other words, the emergent protocol adapts to minimize the information that passes through it.

We will release the code for our experiments upon acceptance.

## 3 METHODOLOGY

### 3.1 GAMES

We study two signaling games. In Guess Number, the agents are trained to recover an integer-representing vector with uniform Bernoulli-distributed components. This simple setup gives us full control over the amount of information needed to solve the task. The second game, Image Classification, uses more naturalistic data, as the agents are jointly trained to classify pairs of hand-written MNIST digits (LeCun et al., 1998b).

**Guess Number** We draw an 8-bit integer $z$, $0 \leq z \leq 255$ uniformly at random, by sampling its 8 bits independently from the uniform Bernoulli distribution. All bits are revealed to Sender as a 8-dimensional binary vector $\boldsymbol{i}_s$. The last $k$ bits are revealed to Receiver ($0 \leq k \leq 8$) as its input $\boldsymbol{i}_r$. Sender outputs a single-symbol message $\boldsymbol{m}$ to Receiver. In turn, Receiver outputs a vector $\boldsymbol{o}$ that recovers all the bits of $z$ and should be equal to $\boldsymbol{i}_s$.

In this game, Sender has a linear layer that maps the input vector $\boldsymbol{i}_s$ to a hidden representation of size 10, followed by a leaky ReLU activation. Next is a linear layer followed by a softmax over the vocabulary. Receiver linearly maps both its input $\boldsymbol{i}_r$ and the message to 10-dimensional vectors, concatenates them, applies a fully connected layer with output size 20, followed by a leaky ReLU. Finally, another linear layer and a sigmoid nonlinearity are applied. When training with REINFORCE and the Stochastic Computation graph approach (see Section 3.2), we increase the hidden layer sizes threefold, as this leads to more robust convergence.

**Image Classification** In this game, the agents are jointly trained to classify 28x56 images of two MNIST digits, stacked side-by-side (more details in Appendix). Unlike Guess Number, Receiver has no side input. Instead, we control the informational complexity of Receiver's task by controlling the size of its output space, i.e., the number of labels we assign to the images. To do so, we group all two-digit sequences 00..99 into $N_l \in \{2, 4, 10, 20, 25, 50, 100\}$ equally-sized classes.

In Sender, input images are embedded a LeNet-1 instance (LeCun et al., 1990) into 400-dimensional vectors. These embedded vectors are passed to a fully connected layer, followed by a softmax selecting a vocabulary symbol. Receiver embeds the received messages into 400-dimensional vectors, passed to a fully connected layer with a softmax activation returning the class probabilities.

We report hyperparameter grids in Appendix. In the following experiments, we fix vocabulary to 1024 symbols (experiments with other vocabulary sizes, multi-symbol messages, and larger architectures are reported in Appendix). No parts of the agents are pre-trained or shared. The optimized loss depends on the gradient estimation method used (see Section 3.2). We denote it $\mathcal{L}(\boldsymbol{o}, \boldsymbol{l})$, and it is a function of Receiver's output $\boldsymbol{o}$ and the ground-truth output $\boldsymbol{l}$. When training in Guess Number with REINFORCE, we use a 0/1 loss: the agents get 0 only if all bits of $z$ were correctly recovered. When training with Gumbel-Softmax relaxation or the Stochastic Computation Graph approach, we use binary cross-entropy (Guess Number) and negative log-likelihood (Image Classification).

## 3.2 TRAINING WITH DISCRETE CHANNEL

Training to communicate with discrete messages is non-trivial, as we cannot back-propagate through the messages. Current language emergence work mostly uses Gumbel-Softmax relaxation (e.g. (Havrylov & Titov, 2017)) or REINFORCE (e.g. (Lazaridou et al., 2016)) to get gradient estimates. We also explore the Stochastic Computation Graph optimization approach. We plug the obtained gradient estimates into the Adam optimizer (Kingma & Ba, 2014).

**Gumbel-Softmax relaxation** Samples from the Gumbel-Softmax (Maddison et al., 2016; Jang et al., 2016) distribution (a) are reperameterizable, hence allow gradient-based training, and (b) approximate samples from the corresponding Categorical distribution. To get a sample that approximates an $n$-dimensional Categorical distribution with probabilities $p_i$, we draw $n$ i.i.d. samples $g_i$ from Gumbel(0,1) and use them to calculate a vector $\boldsymbol{y}$ with components:

$$y_i = \frac{exp\left[(g_i + \log \ p_i)/\tau\right]}{\sum_j exp\left[(g_j + \log \ p_j)/\tau\right]} \tag{2}$$

where $\tau$ is the temperature hyperparameter. As $\tau$ tends to 0, the samples $\boldsymbol{y}$ get closer to one-hot samples; as $\tau \to +\infty$, the components $y_i$ become uniform. During training, we use these relaxed samples as messages from Sender, making the entire Sender/Receiver setup differentiable.

**REINFORCE** by Williams (1992) is a standard reinforcement learning algorithm. In our setup, it estimates the gradient of the expectation of the loss $\mathcal{L}(\boldsymbol{o}, \boldsymbol{l})$ w.r.t. the parameter vector $\boldsymbol{\theta}$ as follows:

$$\mathbb{E}_{\boldsymbol{i}_s, \boldsymbol{i}_r} \mathbb{E}_{\boldsymbol{m} \sim S(\boldsymbol{i}_s), \boldsymbol{o} \sim R(\boldsymbol{m}, \boldsymbol{i}_r)} \left[(\mathcal{L}(\boldsymbol{o}; \boldsymbol{l}) - b)\nabla_{\boldsymbol{\theta}} \log P_{\boldsymbol{\theta}}(\boldsymbol{m}, \boldsymbol{o})\right] \tag{3}$$

The expectations are estimated by sampling $\boldsymbol{m}$ from Sender and, after that, sampling $\boldsymbol{o}$ from Receiver. We use the running mean baseline $b$ (Greensmith et al., 2004; Williams, 1992) as a control variate. We adopt the common trick to add an entropy regularization term (Williams & Peng, 1991; Mnih et al., 2016) that favors higher entropy. We impose entropy regularization on the outputs of the agents with coefficients $\lambda_s$ (Sender) and $\lambda_r$ (Receiver).

**Stochastic Computation Graph** In our setup, the gradient estimate approach of Schulman et al. (2015) reduces to computing the gradient of the following surrogate function:

$$\mathbb{E}_{\boldsymbol{i}_s, \boldsymbol{i}_r} \mathbb{E}_{\boldsymbol{m} \sim S(\boldsymbol{i}_s)} \left[\mathcal{L}(\boldsymbol{o}; \boldsymbol{l}) + stop\_gradient\left(\mathcal{L}(\boldsymbol{o}; \boldsymbol{l}) - b\right) \log P_{\boldsymbol{\theta}}(\boldsymbol{m})\right] \tag{4}$$

Here, we do not sample Receiver actions: Its parameter gradients are obtained with standard back-propagation (the first term in Eq. 4). Sender's messages are sampled, and its gradient are calculated akin to REINFORCE (the second term in Eq. 4). As in REINFORCE, we apply entropy-favoring regularization on Sender's output (with coefficient $\lambda_s$) and use the mean baseline $b$.

**Role of entropy regularization** As we mentioned above, when training with REINFORCE and Stochastic Computation Graph, we include a (standard) entropy regularization term in the loss which

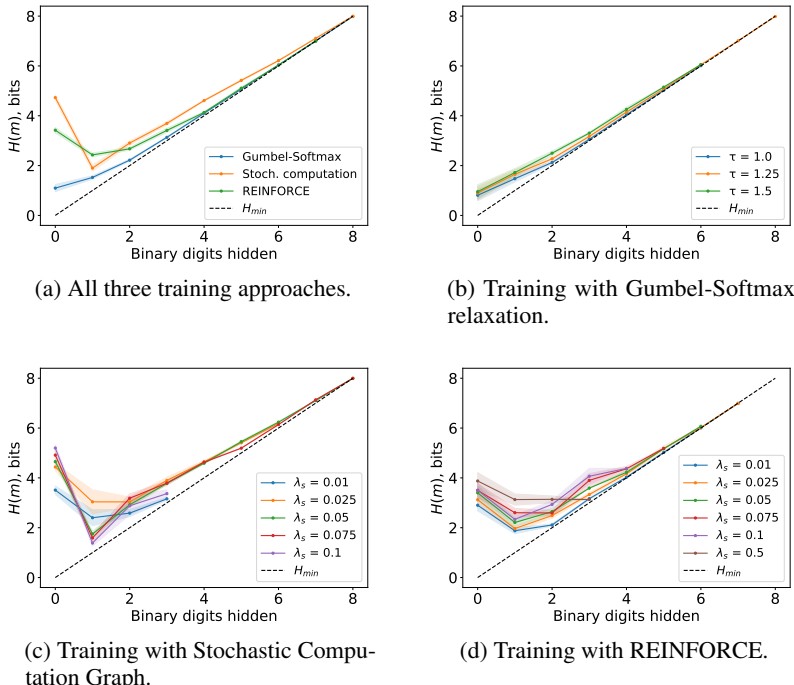

Figure 1: Guess Number: entropy of the messages $\boldsymbol{m}$. Shaded regions mark standard deviation.

explicitly *maximizes* entropy of Sender's output. Clearly, this term is at odds with the entropy *minimization* effect we observe. In our experiments, we found that high values of $\lambda_s$ prevent communication success; on the other hand, small non-zero $\lambda_s$ is crucial for successful training. In Section 4 we investigate the effect of $\lambda_s$ on entropy minimization.

### 3.3 Experimental protocol

In Guess Number, we use all $2^8$ possible inputs for training, early stopping and analysis. In Image Classification, we train on random image pairs from the MNIST training data, and use image pairs from the MNIST held-out set for validation. We select the runs that achieved a high level of performance (training accuracy above 0.99 for Guess Number and validation accuracy above 0.98 for Image Classification), thus studying typical agent behavior *provided they succeeded at the game*.

At test time, we select the Sender's message symbol greedily, hence the messages are discrete and Sender represents a (deterministic) function $S$ of its input $\boldsymbol{i}_s$, $\boldsymbol{m} = S(\boldsymbol{i})$. Calculating the entropy $H(\boldsymbol{m})$ of the distribution of discrete messages $\boldsymbol{m}$ is straightforward. In Guess Number, we enumerate all 256 possible values of $z$ as inputs, save the messages from Sender and calculate entropy $H(\boldsymbol{m})$. For Image Classification, we sample image pairs from the MNIST hold-out set.

The upper bound on $H(\boldsymbol{m})$ is as follow: $H_{max} = 8$ bits (bounded by $H(\boldsymbol{i}_s)$) in Guess Number, and $H_{max} = 10$ bits (bounded by vocabulary size) in Image Classification. Its lower bound is equal to $H_{min} = H(\boldsymbol{l}|\boldsymbol{i}_r) = 8 - k$ bits for Guess number. In Image Classification, communication can only succeed if $H(\boldsymbol{m})$ is not less than $H(\boldsymbol{l})$, i.e., $H_{min} = H(\boldsymbol{l}) = \log_2 N_l$, with $N_l$ the number of equally-sized classes we split the images into.

## 4 Experiments

### 4.1 Entropy minimization

**Guess Number** In Figure 1, the horizontal axes span the number of bits of $z$ that Receiver lacks, $8 - k$. The vertical axis reports the information content of the protocol, measured by messages entropy

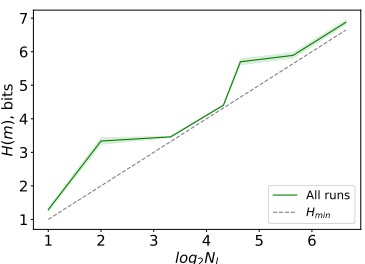
(a) Successful runs pooled together.

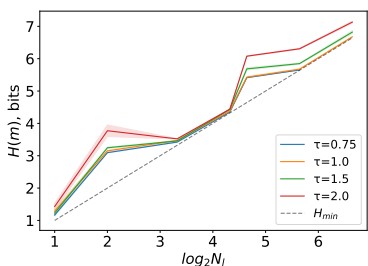
(b) Successful runs grouped by temperature.

Figure 2: Image Classification: entropy of the messages $m$ in function of log number of target classes, $N_l$. Shaded regions mark standard deviation.

$H(m)$. Each integer on the horizontal axis corresponds to a game configuration, and for each such configuration we aggregate multiple (successful) runs with different hyperparameters and random seeds. $H_{min}$ indicates the minimal amount of bits Sender has to send in a particular configuration for the task to be solvable. The upper bound (not shown) is $H_{max} = 8$ bits.

Consider first the configurations where Receiver's input is insufficient to answer correctly (at least one binary digit hidden, $k \leq 7$). From Figure 1a, we observe that the transmitted information is strictly monotonically increasing with the number of binary digits hidden from Receiver. Thus, even if Sender sees the very same input in all configurations, a more nuanced protocol is only developed when it is necessary. Moreover, the entropy $H(m)$ (equivalently: the transmitted information) stays close to the lower bound. This entropy minimization property holds for all the considered training approaches across all configurations.

Consider next the configuration where Receiver is getting the whole integer $z$ as its input ($k = 8$, the leftmost configuration in Figure 1, corresponding to 0 on x axis). Based on the observations above, one would expect that the protocol would approach zero entropy in this case (as no information needs to be transmitted). However, the measurements indicate that the protocol is encoding considerably more information. It turns out that this information is entirely ignored by Receiver. To demonstrate this, we fed all possible distinct inputs to Sender, retrieved the corresponding messages, and *shuffled* them to destroy any information about the inputs they might carry. The shuffled messages were then passed to Receiver alongside with its own (un-shuffled) inputs. The overall performance was not affected by this manipulation, confirming the hypothesis that Receiver ignores messages. We conclude that in this case there is no apparent entropy minimization pressure on Sender simply because there is no communication. The full experiment is reported in Appendix.

We further consider the effect of various hyperparameters. In Figure 1b, we split the results obtained with Gumbel-Softmax by relaxation temperature. As discussed in Section 3.2, lower temperatures more closely approximate discrete communication, hence providing a convenient control of the level of discreteness imposed during training (recall that at test time we select the symbol greedily). The figure shows that lower temperatures consistently lead to lower $H(m)$ values. This implies that, as we increase the "level of discreteness" at training, we get stronger entropy minimization pressures.

In Figures 1c & 1d, we report $H(m)$ when training with Stochastic Graph Optimization and RE-INFORCE across degrees of entropy regularization. We report curves corresponding to $\lambda_s$ values[1] which converged in more than three configurations. With REINFORCE, we see a weak tendency for a higher $\lambda_s$ to trigger higher entropy in the protocol (only violated at $\lambda_s = 0.5$). However, message entropy stays generally close to the lower bound even in presence of strong exploration, which favors higher entropy in Sender's output distribution.

**Image Classification** As the models are more complex, we only had consistent success when training with Gumbel-Softmax. In Figure 2a we aggregate all successful runs. The information encoded by the protocol grows as Receiver's output requires more information. However, in all configurations, the transmitted information stays well below the 10-bit upper bound and tends to be close to $H_{min}$.

---

[1] The parameter $\lambda_r$ that controls the strength of Receiver's entropy regularization turned out not to play role in the observed effect.

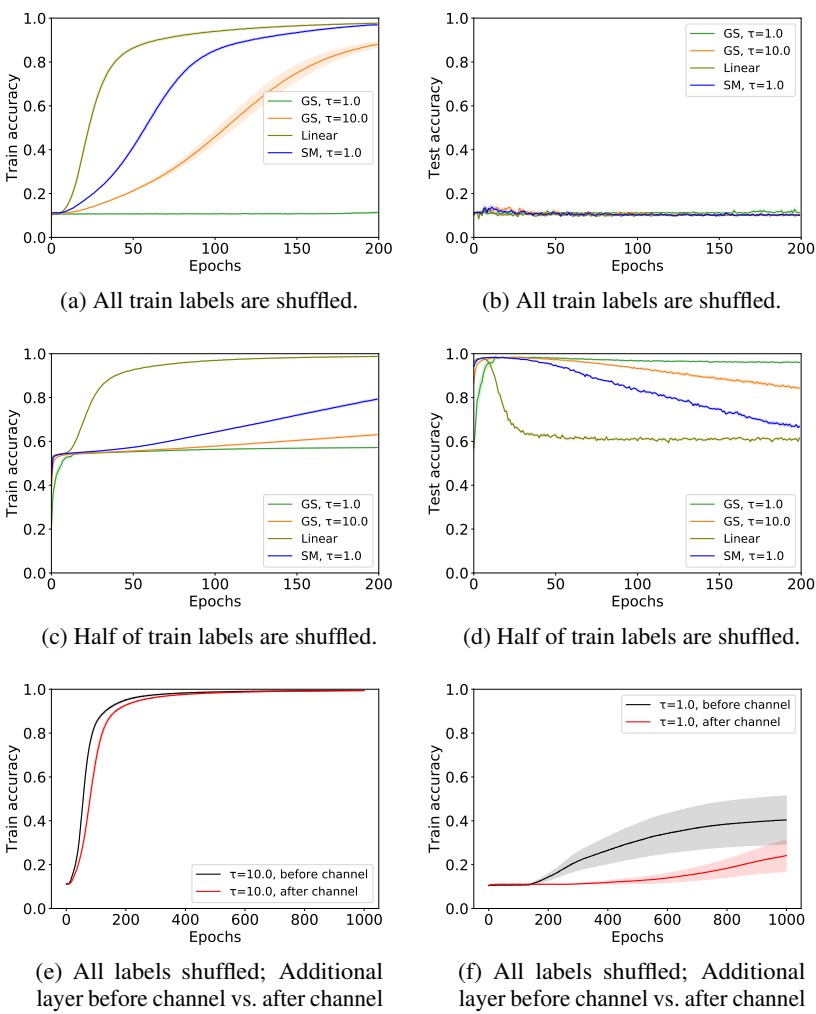

(a) All train labels are shuffled.

(b) All train labels are shuffled.

(c) Half of train labels are shuffled.

(d) Half of train labels are shuffled.

(e) All labels shuffled; Additional layer before channel vs. after channel

(f) All labels shuffled; Additional layer before channel vs. after channel

Figure 3: Learning in presence of random labels. *GS* (*SM*) indicates models trained with Gumbel-Softmax (Softmax) channel. *Linear* are models with the channel removed.

A natural interpretation is that Sender prefers to take charge of image classification and directly pass information about the output label, rather than sending along a presumably more information-heavy description of the input. In Figure 2b, we split the runs by temperature. Again, we see that lower temperatures consistently lead to stronger entropy minimization pressures.

Summarizing, *when communicating through a discrete channel, there is consistent pressure for the emergent protocol to encode as little information as necessary*. This holds across games, training methods and hyperparameters. When training with Gumbel-Softmax, temperature controls the strength of this pressure, confirming the relation between entropy minimization and discreteness.

## 4.2 REPRESENTATION DISCRETENESS AND ROBUSTNESS

The entropy minimization effect we established in Section 4.1 indicates that a discrete representation will only store as much information as necessary to solve the task. This emergent behavior respects the "information bottleneck" principle (Tishby et al., 1999; Achille & Soatto, 2018a). The fact that lower training-time temperatures in Gumbel-Softmax optimization correlate with both higher discreteness and a tighter bottleneck (see Section 4.1) makes us further conjecture that discreteness is causally connected to the emergent bottleneck.

Interestingly, the same information-bottleneck principle has also been claimed to govern entropy minimization in natural language (Zaslavsky et al., 2018; 2019). The information-bottleneck effects in neural agents and natural language might be due to the same cause, namely discreteness of the communication. Further, we hypothesize that the emergent discrete information-bottleneck might have useful properties, since existing (continuous) architectures that explicitly impose a bottleneck pressure to provide a form of beneficial regularization are more robust to overfitting (Fischer, 2019) and adversarial attacks (Alemi et al., 2016; Fischer, 2019). We test here whether the expected regularization properties also emerge in our computational simulations (without any explicit pressure imposed through the cost function), and whether they are correlated with the degree of discreteness of the communication channel. If this connection exists, this also suggests that discreteness might be "beneficial" to human languages for the same reasons.

To assess our hypotheses, we consider the Image Classification game ($N_l = 10$) in presence of randomly-shuffled training labels (the test set is untouched) (Zhang et al., 2016). This task allows us to explore whether the discrete communication bottleneck is associated to robustness to overfitting, and whether the latter depends on the level of discreteness (controlled by the temperature $\tau$ of the Gumbel-Softmax relaxation).[2]

We use the same architecture as before. The agents are trained with Gumbel-Softmax relaxation. However, for practicality, we do not switch to fully discrete communication at test time, only removing the noise component, thus effectively reducing Sender's output to softmax with temperature. We refer to this architecture as GS.

We also consider two baseline architectures without relaxed discrete channel. In Linear, the fully connected output layer of Sender is directly connected to the linear embedding input of Receiver. Softmax (SM) places a softmax activation (with temperature) after Sender's output layer and passes the result to Receiver. At test time, SM coincides with GS with the same temperature, but there was no discrete-sampling approximation during SM training.

We vary temperature and proportion of training examples with shuffled labels. We use temperatures $\tau = 1.0$ and $\tau = 10.0$ (the agents reach a test accuracy of 0.98 when trained with these temperatures on the original training set). SM with $\tau = 1.0$ and $\tau = 10.0$ behave similarly, hence we only report SM with $\tau = 1.0$.

In Figure 3a we report *training* accuracy when all labels are shuffled. Linear and SM fit the random labels almost perfectly within the first 150 epochs. With $\tau = 10.0$, GS achieves 0.8 accuracy within 200 epochs. When GS with $\tau = 1.0$ is considered, the agents only start to improve over random guessing after 150 epochs, and accuracy is well below 0.2 after 200 epochs. As expected, test set performance is at chance level (Figure 3b). In the next experiment, we shuffle labels for a randomly selected half of the training instances. Train and test accuracies are shown in Figures 3c and 3d, respectively. All models initially fit the true-label examples (train accuracy $\approx 0.5$, test accuracy $\approx 0.97$). With more training, the baselines and GS with $\tau = 10.0$ start (over)fitting the randomly labeled examples, too: train accuracy grows, while test accuracy falls. In contrast, GS with $\tau = 1.0$ does not fit random labels, and its test accuracy stays high. Note that SM patterns with Linear and high-temperature GS, showing that the training-time discretization noise in GS is instrumental for robustness to over-fitting.

We interpret the results as follows. To fully exploit their joint capacity for "successful" over-fitting, the agents need to coordinate label memorization. This requires passing large amounts of information through the channel. With a low temperature (more closely approximating a discrete channel), this is hard, due to stronger entropy minimization pressure. To test the hypothesis, we ran an experiment where all labels are shuffled and a layer of size 400x400 is either added to Sender (just before the channel) or to Receiver (just after the channel). We predict that, with higher $\tau$ (less discrete, less entropy minimization pressure), the training curves will be close, as the extra capacity can be used for memorization equally easy in both cases. With lower $\tau$ (better discrete approximation, more pressure), the accuracy curves will be more distant, as the extra capacity can only be successfully exploited for memorization when placed *before* the channel. Figures 3e & 3f borne out the prediction.

---

[2]The appendix reports experiments showing that agents trained with a low-temperature channel also display increased robustness against adversarial attacks.

These experiments showed that increased channel discreteness makes it harder to pass large amounts of information through, and leads to increased robustness against overfitting. This supports our hypotheses that discreteness brings about a bottleneck that in turn has some beneficial properties, which might ultimately provide a motivation for why an emergent communication system should evolve towards discreteness.

## 5 RELATED WORK

We briefly reviewed studies of emergent deep agent communication and entropy minimization in human language in the introduction. We are not aware of earlier work that looks for this property in emergent communication, although Evtimova et al. (2018) used information theory to study protocol development during learning, and, closer to us, Kågebäck et al. (2018) studied the effect of explicitly adding a complexity minimization term to the cost function on an emergent color-naming system.

Discrete representations are explored in many places (e.g., van den Oord et al., 2017; Jang et al., 2016; Rolfe, 2016). However, these works focus on ways to learn discrete representations, rather than analyzing the properties of representations that are independently emerging on the side. Furthermore, our study also extends to the agents communicating with variable-length messages, produced and consumed by GRU (Cho et al., 2014) and Transformer (Vaswani et al., 2017) cells (see Appendix C.3). The sequential setup is specific to language, clearly distinguished from the settings studied in generic sparse-representation work.

Other studies, inspired by the informational bottleneck principle, control the complexity of neural representations by regulating their information content (Strouse & Schwab, 2017; Fischer, 2019; Alemi et al., 2016; Achille & Soatto, 2018a;b). While they externally impose the bottleneck, we observe that it is an intrinsic feature when learning to communicate through a discrete channel.

## 6 DISCUSSION

Entropy minimization is pervasive in human language, where it constitutes a specific facet of the more general pressure towards communication efficiency. We found that the same property consistently characterizes the protocol emerging in simulations where two neural networks learn to solve a task jointly through a discrete communication code.

In a comparative perspective, our results suggest that entropy minimization is a general property of discrete communication systems, independent of specific biological constraints humans are subject to. In particular, our analysis tentatively establishes a link between this property and the inherent difficulty of encoding information in discrete form (cf. the effect of adding a layer before or after the communication bottleneck in the overfitting experiment above).

Exploring entropy minimization in computational simulations provides a flexibility we lack when studying humans. For example, we uncovered here initial evidence that the communication bottleneck is acting as a good regularizer, making the joint agent system more robust to noise. This leads to an intriguing conjecture on the origin of language. Its discrete nature is often traced back to the fact that it allows us to produce an infinite number of expressions by combining a finite set of primitives (e.g., Berwick & Chomsky, 2016). However, it is far from clear that the need to communicate an infinite number of concepts could have provided the initial pressure to develop a discrete code. More probably, *once such code independently emerged*, it laid the conditions to develop an infinitely expressive language (Bickerton, 2014; Collier et al., 2014). Our work suggests that, because of its inherent regularizing effect, discrete coding is advantageous already when communication is about a limited number of concepts, providing an alternative explanation for its origin.

In the future, we would like to study more continuous domains, such as color maps, where perfect accuracy is not easily attainable, nor desirable. Will the networks find an accuracy/complexity trade-off similar to those attested in human languages? Will other core language properties claimed to be related to this trade-off, such as Zipfian frequency distributions (Ferrer i Cancho & Díaz-Guilera, 2007), concurrently emerge? We would also like to compare the performance of human subjects equipped with novel continuous vs. discrete communication protocols, adopting the methods of experimental semiotics (Galantucci, 2009). We expect discrete protocols to favor generalization and robustness.

Our results have implications for the efforts to evolve agents interacting with each other and with humans through a discrete channel. First, because of entropy minimization, we should not expect the agents to develop a richer protocol than the simplest one that will ensure accurate communication. For example, Bouchacourt & Baroni (2018) found that agents trained to discriminate pairs of natural images depicting instances of about 500 high-level categories, such as cats and dogs, developed a lexicon that does not denote such categories, but low-level properties of the image themselves. This makes sense from an entropy-minimization perspective, as talking about the 500 high-level categories demands $\log_2 500$ bits of information, whereas many low-level strategies (e.g., discriminating average pixel intensity in the images) will only require transmitting a few bits. To have agents developing rich linguistic protocols, we must face them with varied challenges that truly demand them.

Second, the focus on a discrete protocol is typically motivated by the goal to develop machines eventually able to communicate with humans. Indeed, discrete messages are not required in multi-agent scenarios where no human in the loop is foreseen (Sukhbaatar et al., 2016). Our results suggest that, long before agents reach the level of complexity necessary to converse with humans, there are independent reasons to encourage discreteness, as it provides a source of robustness in a noisy world. An exciting direction for future applied work will be to test, in more practical settings, the effectiveness of discrete communication as a general form of representation learning.

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

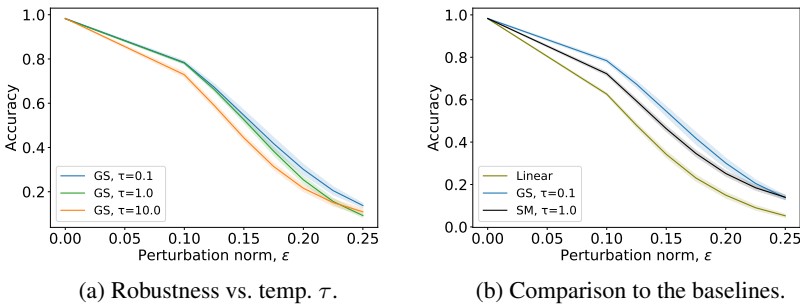

(a) Robustness vs. temp. $\tau$.  (b) Comparison to the baselines.

Figure 4: Robustness to adversarial examples: higher accuracy given fixed $\epsilon$ implies more robustness.

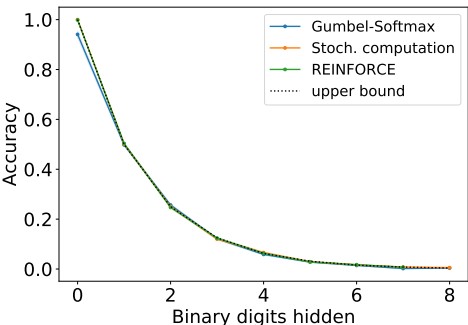

Figure 5: Guess Number: Receiver's dependence on messages, measured as performance drop under message intervention.

## A  ROBUSTNESS TO ADVERSARIAL ATTACKS

In this Section, we study robustness of the agents equipped with a relaxed discrete channel against adversarial attacks. We use the same architectures as in Section 4.2 of the main paper. Specifically, by GS we indicate the architecture where agents are trained with Gumbel-Softmax relaxation, which at test-time is replaced by (noiseless) softmax with the same temperature. SM is an architecture where the communication channel is replaced by a Softmax layer with temperature. The Linear baseline has no "channel": the output of Sender is directly plugged as input to the Receiver.

We train agents with different random seeds and implement white-box attacks on the trained models, varying temperature $\tau$ and the allowed perturbation norm, $\epsilon$. We use the standard *Fast Gradient Sign Method* (FGSM) of Goodfellow et al. (2014). The original image $\boldsymbol{i}_s$ is perturbed to $\boldsymbol{i}_s^*$ along the direction that maximizes the loss of Receiver's output $\boldsymbol{o} = R(S(\boldsymbol{i}_s))$ w.r.t. ground-truth class $\boldsymbol{l}$:

$$\boldsymbol{i}_s^* = clip\left[\boldsymbol{i}_s + \epsilon \cdot sign\left[\nabla_{\boldsymbol{i}_s}\mathcal{L}(\boldsymbol{o}, \boldsymbol{l})\right], 0, 1\right] \tag{5}$$

where $\epsilon$ controls the $L_\infty$ norm of the perturbation. Under an attack with a fixed $\epsilon$, a more robust method would have a smaller accuracy drop. To avoid numerical stability issues akin to those reported by Carlini & Wagner (2016), all computations are done in 64-bit floats.

As earlier in Section 4.2, we observed that SM behaves similarly with different temperatures (we experimented with $\tau \in \{0.1, 1.0, 10.0\}$), we report only results with $\tau = 1.0$. Figure 4a shows that, as the relaxation temperature decreases, the accuracy drop also decreases. The highest robustness is achieved with $\tau = 0.1$. Comparison with the baselines (Figure 4b) confirms that relaxed discrete training with $\tau = 0.1$ improves robustness. However, this robustness comes at the cost of harder training: 2 out of 5 random seeds did not reach the desired performance level (0.98) after 200 epochs.

## B    HOW MUCH DOES RECEIVER RELY ON MESSAGES IN GUESS NUMBER?

We supplement the experiments of Section 3 of the main text by studying the degree to which Receiver relies on messages in Guess Number. In particular, we show that when Receiver has the full input ($\boldsymbol{i}_s = \boldsymbol{i}_r$), it ignores the messages.

We measure the degree to which Receiver relies on the messages from Sender by constructing a setup where we break communication, but still let Receiver rely on its own input. More precisely, we first enumerate all test inputs for Sender $\boldsymbol{i}_s$ and Receiver $\boldsymbol{i}_r$. We obtain messages that correspond to Sender's inputs, and shuffle them. Next, we feed the shuffled messages alongside Receiver's own (un-shuffled) inputs and compute accuracy, as a measure of Receiver's dependence on the messages. This procedure preserves the marginal distribution of the Receiver input messages, but destroys all the information Sender transmits.

Without messages, Receiver, given $k$ input bits, can only reach an accuracy of $2^{8-k}$. In Figure 5, we report results aggregated by training method. Receiver is extremely close to the accuracy's higher bound in all configurations. Moreover, when Receiver gets the entire input, the drop in accuracy after shuffling is tiny, proving that Receiver's reliance on the message is minimal in that setting.

## C    INFLUENCE OF ARCHITECTURE CHOICES

### C.1    DOES VOCABULARY SIZE AFFECT THE RESULTS?

We repeat the same experiments as in Section 3 of the main text while varying vocabulary size. Note that, to make Guess Number solvable across each configuration, the vocabulary has to contain at least 256 symbols. Similarly, for Image Classification, vocabulary size must be of at least 100. We tried vocabulary sizes of 256, 1024, 4096 for Guess Number, and 512, 1024, 2048 for Image Classification. The results are reported in Figures 6 (Guess Number) and 7 (Image Classification). We observe that there is little qualitative variation over vocabulary size, hence the conclusions we had in Section 3 of the main paper are robust to variations of this parameter.

### C.2    DOES RECEIVER'S CAPACITY AFFECT THE RESULTS?

One potential confounding variable is the capacity of Receiver. Indeed, if Receiver is very simple, then, for the task to be solved, Sender would have to calculate the answer itself and feed it to Receiver. To investigate this, we repeat the Image Classification experiment from Section 4 of the main paper while controlling the power of Receiver's architecture: we put two additional fully-connected 400x400 hidden layers between the input embedding and the output layer, while in Section 4, Receiver had a single hidden layer.

In Figure 8 we compare the results obtained with these two variations of Receiver. The reported entropy minimization effect holds: even in presence of additional layers, the entropy of messages $H(\boldsymbol{m})$ is far from the upper-bound $H_{max} = 10$ bits and closely follows the lower bound, $H_{min} = \log_2 N_l$. Thus, again, a more nuanced protocol only appears when it is needed. Finally, we see that results for both architectures are close, although in three out of seven task setups (the number of classes $N_l$ is 2, 10, and 20) a deeper model results in a slightly higher entropy of the protocol, on average. Overall, we conclude that Receiver's capacity does not play a major role in the entropy minimization effect and the latter also takes place with a more powerful Receiver.

### C.3    WHAT IF COMMUNICATION TAKES PLACE THROUGH SEQUENCES OF SYMBOLS?

We also experiment with Guess Number in a setup where the agents communicate via variable-length messages. The general architecture of the agents is same as in Section 3.1, however the output of Sender is used as the initial hidden state of a GRU cell (Cho et al., 2014). In turn, this GRU is unrolled to generate the message. The message is produced until the GRU outputs a special *<eos>* token or until the maximal length is reached. In the latter case, *<eos>* is appended to the message. The produced message is consumed by a Receiver's GRU unit and the hidden state corresponding to *<eos>* is used by Receiver as input to further processing. We use the Stochastic Computation Graph estimator as described in Section 3.2, as it provided fastest convergence.

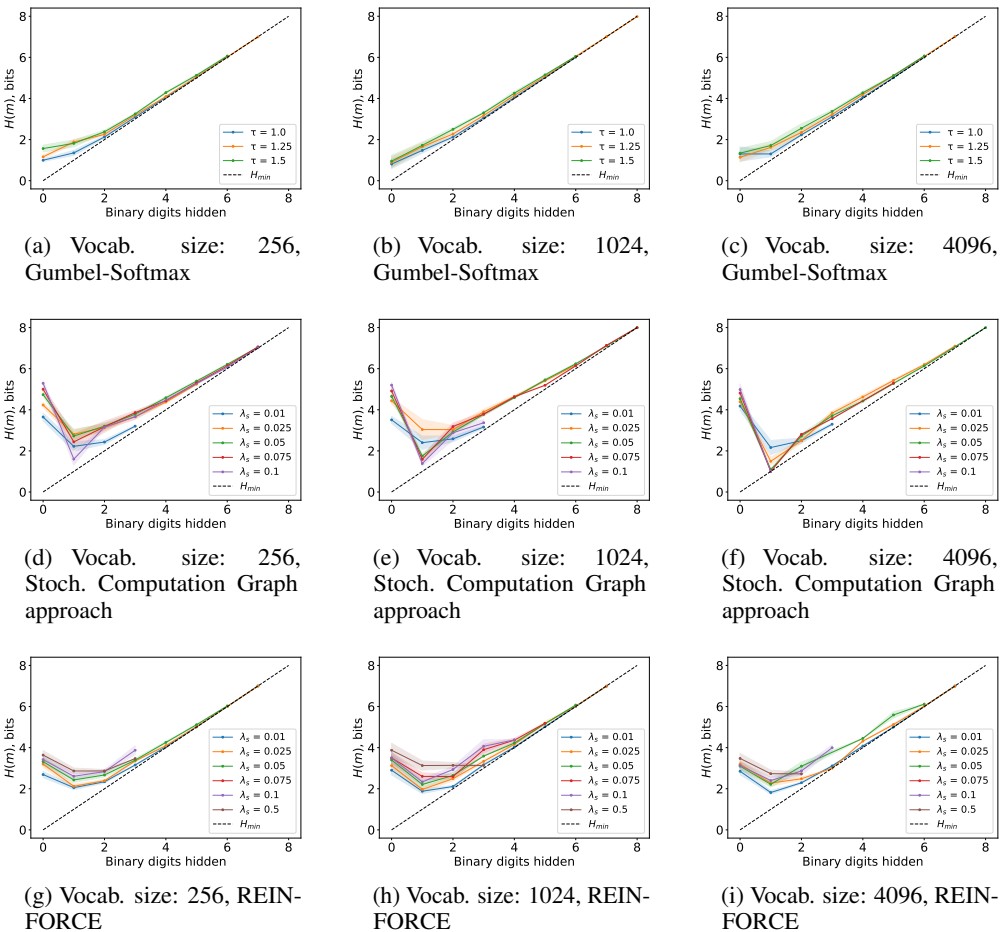

(a) Vocab. size: 256, Gumbel-Softmax

(b) Vocab. size: 1024, Gumbel-Softmax

(c) Vocab. size: 4096, Gumbel-Softmax

(d) Vocab. size: 256, Stoch. Computation Graph approach

(e) Vocab. size: 1024, Stoch. Computation Graph approach

(f) Vocab. size: 4096, Stoch. Computation Graph approach

(g) Vocab. size: 256, REIN-FORCE

(h) Vocab. size: 1024, REIN-FORCE

(i) Vocab. size: 4096, REIN-FORCE

Figure 6: Guess Number: Entropy of the messages $m$, depending on vocabulary size, training method, and relaxation temperature $\tau$ (when trained with Gumbel-Softmax) or Sender's entropy regularization coefficient $\lambda_s$. Shaded regions mark standard deviation.

We consider the entire variable-length message as the realization of a random variable $m$ when calculating the entropy of the messages, $H(m)$. The results are reported in Figure 9, arranged in function of maximal message length and vocabulary size. As before, we aggregate the successful runs according to the entropy regularization coefficient $\lambda_s$ applied to Sender's output layer.

From Figure 9 we observe that the results are in line with those obtained in the one-symbol scenario. Entropy minimization still holds: a more nuanced (high-entropy) protocol only develops when more digits are hidden from Receiver, which hence requires more information to perform the task. The approximation to the lower bound is however less tight as the overall number of possible messages grows (higher maximum length and/or vocabulary size). There is also a weak tendency for lower $\lambda_s$ to encourage a tighter bottleneck.

In preliminary experiments, we have similar results when the variable-length communication is performed via Transformer cells (Vaswani et al., 2017) instead of GRUs (not reported here).

## D    TWO-DIGIT MNIST DATASET

As discussed in Section 3, to ensure high output informational complexity in the Image Classification task, we use a two-digit variant of the MNIST dataset (LeCun et al., 1998a). We construct it as follows. When iterating over the original MNIST dataset, we take a batch $b$ and (a) select the first

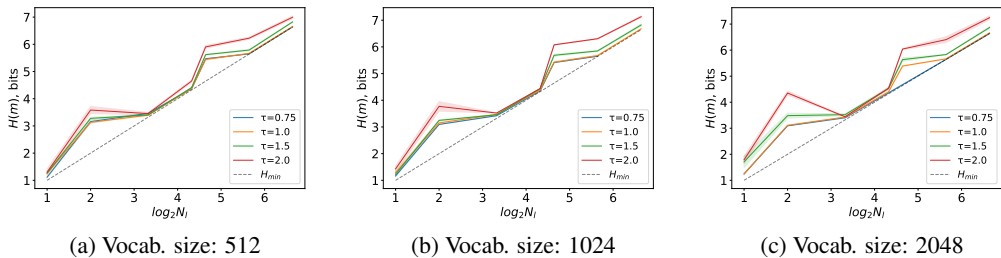

Figure 7: Image Classification: entropy of the messages $H(\boldsymbol{m})$ across vocabulary sizes. Successful runs are pooled together. Shaded regions mark standard deviation.

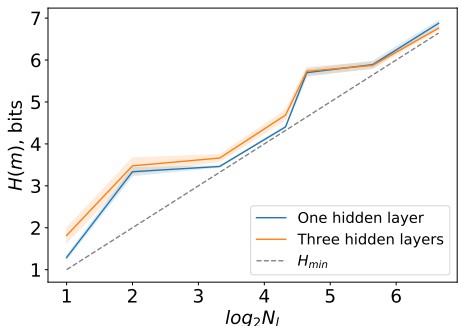

Figure 8: Image Classification: entropy of the messages $H(\boldsymbol{m})$ across Receiver model sizes. Successful runs are pooled together. Shaded regions mark standard deviation.

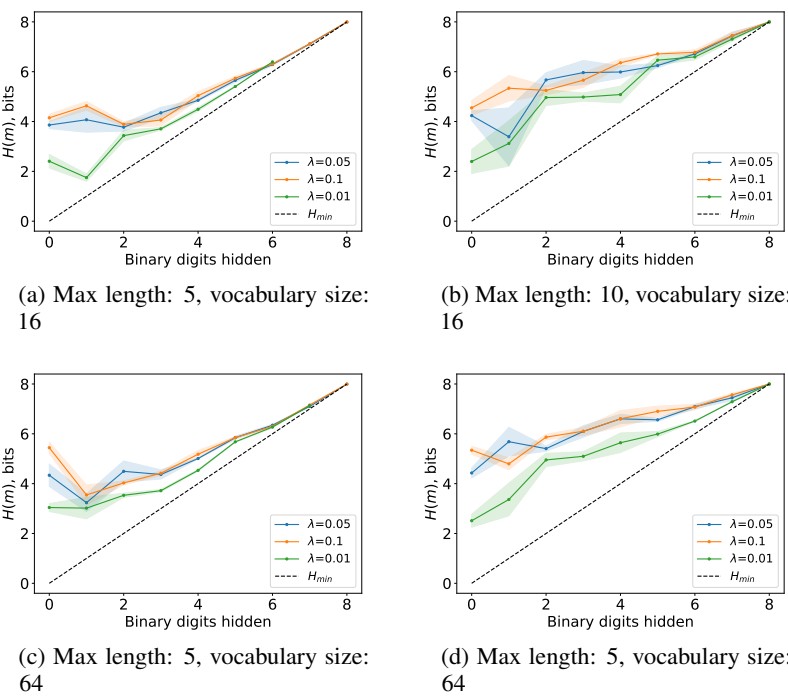

Figure 9: Guess Number: Entropy of the emergent protocol when communication is performed with variable-length messages. Shaded regions mark standard deviation.

$|b|/2$ and last $|b|/2$ images, refer to them as $b_1$ and $b_2$, respectively; (b) create a new batch where the $i$th image from $b_1$ is placed to the left of the $i$th image from $b_2$ and then *vice versa*. As a result, we obtain a new stream of images, where each MNIST digit is seen twice, on the left and on the right side. Note that not all possible pairwise combinations of the original images are generated (there are $60000^2$ of those in the training set alone) and the exact combinations change across epochs. As labels, we use the depicted two-digit number modulo $N_l$, where $N_l$ is the required number of classes. All pixels are scaled into [0, 1]. We use the same process to generate training and test sets, based on the training and test images of the original MNIST dataset, respectively.

## E    HYPERPARAMETERS

In our experiments, we used the following hyperparameter grids.

**Guess Number (Gumbel-Softmax)** Vocab. size: [256, 1024, 4096]; temperature, $\tau$: [0.5, 0.75, 1.0, 1.25, 1.5]; learning rate: [0.001, 0.0001]; max. number of epochs: 250; random seeds: [0, 1, 2, 3]; batch size: 8; early stopping thr.: 0.99; bits shown to Receiver: [0, 1, 2, 3, 4, 5, 6, 7, 8].

**Guess Number (REINFORCE)** Vocab. size: [256, 1024, 4096]; Sender entropy regularization coef., $\lambda_s$: [0.01, 0.05, 0.025, 0.1, 0.5, 1.0]; Receiver entropy regularization coef., $\lambda_r$: [0.01, 0.1, 0.5, 1.0]; learning rate: [0.0001, 0.001, 0.01]; max. number of epochs: 1000; random seeds: [0, 1, 2, 3]; batch size: 2048; early stopping thr.: 0.99; bits shown to Receiver: [0, 1, 2, 3, 4, 5, 6, 7, 8].

**Guess Number (Stochastic Computation Graph approach)**: Vocab. size: [256, 1024, 4096]; Sender entropy regularization coef., $\lambda_s$: [0.01, 0.025, 0.05, 0.075, 0.1, 0.25]; learning rate: [0.0001, 0.001]; max. number of epochs: 1000; random seeds: [0, 1, 2, 3]; batch size: 2048; early stopping thr.: 0.99; bits shown to Receiver: [0, 1, 2, 3, 4, 5, 6, 7, 8].

**Image Classification experiments** Vocab. size: [512, 1024, 2048]; temperature, $\tau$: [0.5, 0.75, 1.0, 1.5, 2.0]; learning rate: [0.01, 0.001, 0.0001], max. number of epochs: 100; random seeds: [0, 1, 2]; batch size: 32; early stopping thr.: 0.98; number of classes: [2, 4, 10, 20, 25, 50, 100].

**Fitting random labels experiments** Vocab. size: 1024; temperature, $\tau$: [1.0, 10.0]; learning rate: 1e-4, max. number of epochs: 200; random seeds: [0, 1, 2, 3, 4]; batch size: 32; early stopping thr.: $\infty$; prob. of label corruption: [0.0, 0.5, 1.0].

**Adversarial attack experiments** Vocab. size: 1024; temperature, $\tau$: [0.1, 1.0, 10.0]; learning rate: 1e-4, max. number of epochs: 200; random seeds: [0, 1, 2, 3, 4]; batch size: 32; early stopping thr.: 0.98.

## F    EVOLUTION OF MESSAGE ENTROPY DURING TRAINING

In this Section, we aim to gain additional insight into development of the communication protocol by measuring its entropy during training. We concentrate on Guess Number and use the same experimental runs summarized in Figure 1 of the main text.

For each game configuration (that is, number of bits hidden from Receiver), we randomly select one successful run and plot the evolution of Sender message entropy and accuracy over training epochs.[3] We also plot entropy and accuracy curves for a randomly selected failed run, to verify to what extent entropy development depends on task success.

We report results for runs where training was performed with Gumbel-Softmax relaxation and with the Stochastic Graph Computation approach in Figures 10 and 11, respectively. The reported entropy and accuracy values are calculated in evaluation mode, where Sender's output is selected greedily, without sampling. A higher entropy of such deterministic Sender indicates that the latter can encode more information about inputs in its messages.

From these results, we firstly observe that the initial entropy of Sender's messages (before training) can be both higher than required for communication success (Figures 10a and 11a) and lower (the rest). When it starts higher than needed, it generally falls closer to the minimum level required for the

---

[3]We exclude the configuration in which Receiver sees the entire input, as it is a degenerate case of non-communication, as discussed in Section 4.

solution. When the initial value is low, it increases during training. The failed runs can have message entropy above (Figures 10a, 10b & 11a) and below (e.g. Figures 10c, 10d & 11d) successful runs, suggesting that there is no systematic relation between degree of entropy and task success.

The fact that the entropy can be reduced with no decrease in accuracy or even with accuracy growth (e.g. Figure 10a, red line, epochs 5..30) indicates that the tendency to discover new messages (increasing entropy) is counter-balanced by the complexity of mutual coordination with Receiver when entropy is larger. In our interpretation, it is this interplay that serves as a source of the natural bottleneck.

Finally, while in some runs the entropy is effectively increased w.r.t. its initialization level, the resulting protocol's entropy is at, or slightly above the lower bound of what the task allows. In this sense, we argue that the reported effect can be correctly denoted as a "minimization" result.

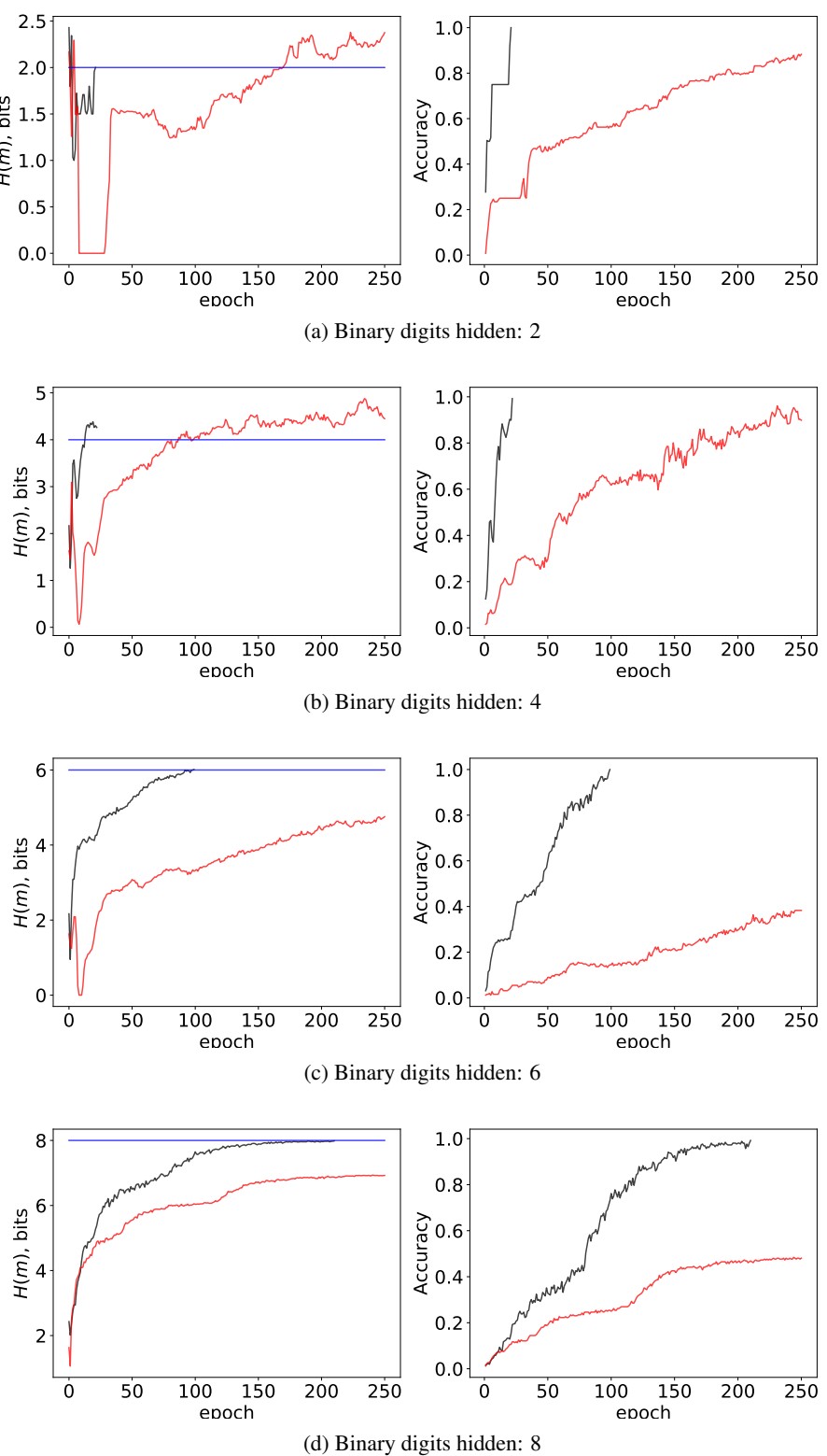

(a) Binary digits hidden: 2

(b) Binary digits hidden: 4

(c) Binary digits hidden: 6

(d) Binary digits hidden: 8

Figure 10: Evolution of $H(m)$ over training epochs. Gumbel Softmax-based optimization, Guess Number. For each game configuration, specified by the number of bits Receiver lacks, we sample one successful (black line) and one failed (red line) training trajectory. The blue line marks $H_{min}$, minimal entropy for a successful solution.

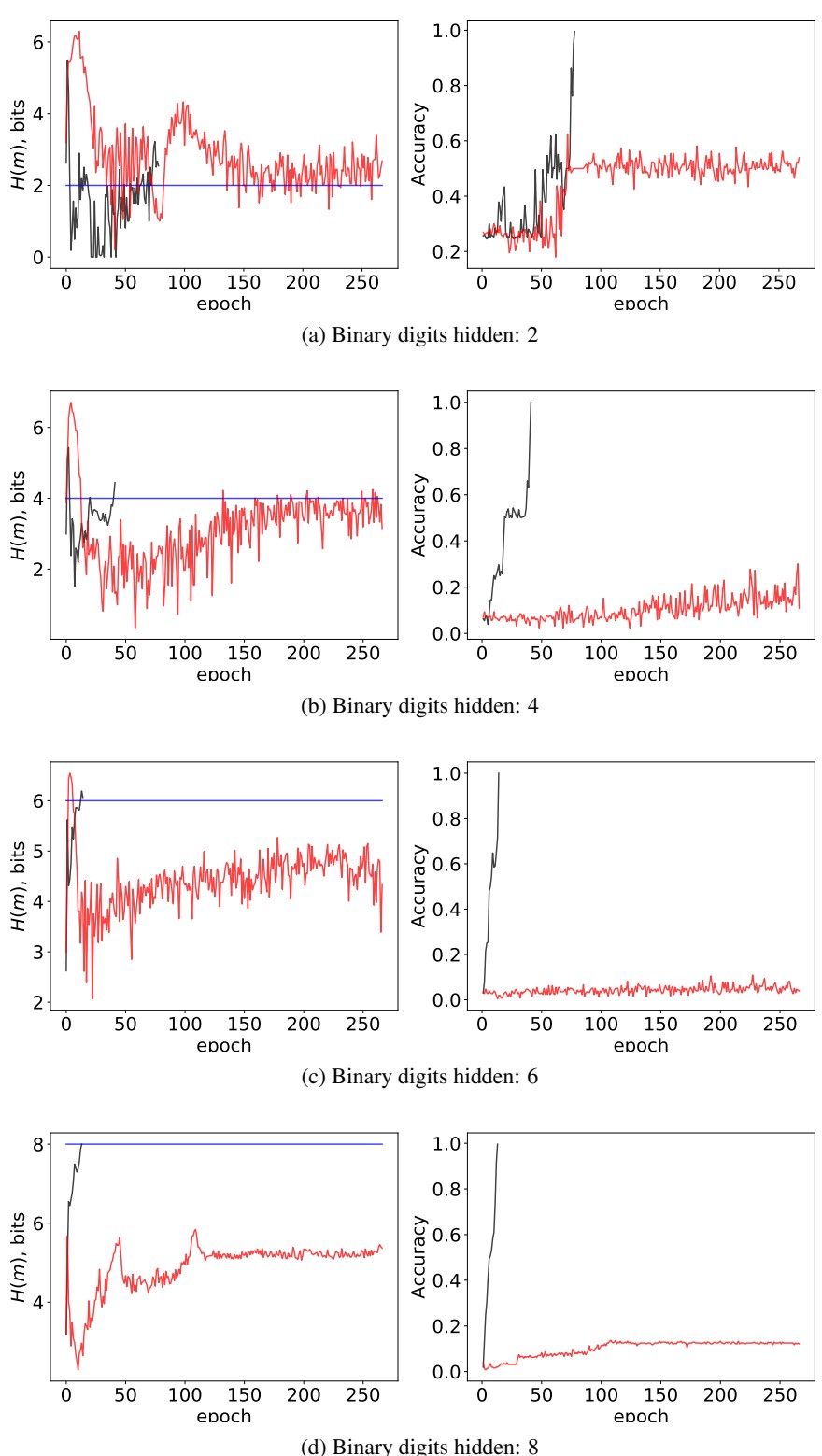

Figure 11: Evolution of $H(m)$ over training epochs. Stochastic Computation Graph-based optimization, Guess Number. For each game configuration, specified by the number of bits Receiver lacks, we sample one successful (black line) and one failed (red line) training trajectory. The blue line marks $H_{min}$, minimal entropy for a successful solution.

