# OpenReview forum: "Entropy Minimization In Emergent Languages"
_ICLR.cc/2020/Conference — Reject_

### Official Review · AnonReviewer1 · 2019-10-23
**Official Blind Review #1**

**Rating:** 6

**Review:**

The authors present a series of simple experiments to characterize the objective that emergent languages are optimizing and why we see various behaviors (not aligned with natural language) when training them to play these language games.  The paper is clearly written, the tasks are simple and minimalist (in a good way) and the experiments are often followed by the exact ablation I was just noting that I wanted to see.

Questions:
1. What happens if the vocabulary is too small to completely communicate the space?  Do models accept that some concepts are not expressible, discover a random behavior that overloads a lexical item, or fail to learn entirely?

2. Why are values for lamda_r not investigated or plotted?

3. How many runs are averaged for each experiment? Should I assume one per seed? why are there different numbers of seeds for each experiment?

4. Page 5 -- "We conclude that in this case there is no apparent entropy minimization pressure on Sender simply because there is no communication" -- Was any analysis performed on the gradients to see if they are completely random or what kind of signal they are getting?

5. How important is the architectural design to these experiments? e.g. number of hidden layers, etc?

**Experience Assessment:**

I have read many papers in this area.

**Review Assessment: Checking Correctness Of Derivations And Theory:**

I assessed the sensibility of the derivations and theory.

**Review Assessment: Checking Correctness Of Experiments:**

I assessed the sensibility of the experiments.

**Review Assessment: Thoroughness In Paper Reading:**

I read the paper at least twice and used my best judgement in assessing the paper.

---

> ### Author Response · Authors · 2019-11-07
> **Answer to AnonReviewer1**
>
> Thank you for your feedback and we are happy you like our paper! Please find our clarifications below.
>
> (a) "What happens if the vocabulary is too small to completely communicate the space?  Do models accept that some concepts are not expressible, discover a random behavior that overloads a lexical item, or fail to learn entirely?"
>
> Thank you for an interesting suggestion! Note that in our reported setup we, at *evaluation* time, do not sample from Sender's output, hence random behavior is not possible.
> Following your suggestion, we have now conducted a small-scale study on Guess Number. It seems that (1) the agents can reach the maximal accuracy allowed by the configuration (i.e. $2^{-(8 - k - log_2 vocab~size)}$), (2) the protocol can reach the upper-bound on the allowed number of bits ($log_2 vocab~size$).
> Some manual inspection indeed shows that some concepts become impossible to express, e.g., when Receiver lacks 3 bits (n-k=3) and Sender can only use a vocabulary of size 2, each of the symbols would encode some configuration of the missing bits (symbol 1 encodes '000' and symbol 0 encodes '111', while eg '001' is not expressible).
>
> (b) "Why are values for lamda_r not investigated or plotted?"
>
> We vary $\lambda_r$ during the hyperparameter grid-search (Appendix E). However, we found it not to influence the studied effect. Since this coefficient is applied to the entropy of Receiver's output layer, we believe this finding is intuitive. Hence, we decided to exclude it from the figures to save space and have a focused discussion. We will mention this in the text.
>
> (c) "How many runs are averaged for each experiment? Should I assume one per seed?"
>
> To obtain Figures 1 & 2, we pooled together all runs that converged during the grid-search. More precisely, in the experiments in Figure 1, 50% of all runs converged for GS and Stochastic Computational Graph approach, and 20% of all runs converged for REINFORCE. In the experiment reported in Figure 2 (GS only), 75% of runs converged.
> In Figure 3 we used all runs from the hyper-parameter grid.
>
> (d) "why are there different numbers of seeds for each experiment?"
> To reduce the variance in the results, we used more seeds in the experiments where it was not prohibitive due to the number of other hyperparameters or training time.
>
> (e) "Page 5 -- "We conclude that in this case there is no apparent entropy minimization pressure on Sender simply because there is no communication" -- Was any analysis performed on the gradients to see if they are completely random or what kind of signal they are getting?"
>
> We have not performed this analysis and we are not sure if it will provide any further information.
>
> (f) "How important is the architectural design to these experiments? e.g. number of hidden layers, etc?"
>
> In Appendix C, we investigate how the potential design choices affect the results: we vary vocabulary size, capacity of Receiver, and experiment with multi-symbol messages, produced and consumed by GRU cells. We have also performed experiments with Transformers, but decided not to include the results to maintain the paper reasonably sized. In all these configurations, we observe the entropy minimization pressure on the emergent protocol.

---

### Official Review · AnonReviewer3 · 2019-10-23
**Official Blind Review #3**

**Rating:** 6

**Review:**

This paper investigates the phenomenon of entropy minimization in emergent languages. The paper studies two simple speaker-listener prediction tasks where the amount of information that’s needed to be encoded in the speaker’s message can be manipulated. They find, in both games and across training methods, that speaker agents generally learn to minimize the amount of information conveyed in their messages (i.e. minimize the entropy), such that they are still able to solve the task. The paper then conducts some experiments measuring the robustness to overfitting, and find that more discrete channels leads to better generalization when training on partially shuffled labels.

Overall, this paper is very well written. I think the main result, that speaker agents learn to only convey as much information as is needed to solve the task, is interesting and insightful. I do, however, have a slight concern about the novelty of this result. Given that the environments are very simple (consisting of a single message sent by a speaker and a prediction made by a listener), the line between ‘multi-agent emergent communication’ and ‘neural network with discrete representations’ is very blurred. This is addressed only briefly in the related work section, with a handful of post-2016 papers cited. I’d personally want to see a more expanded related work section that goes into some more depth into some of these papers, to be able to better judge the novelty of this contribution.

I found the second result, that making the representations more discrete (by lowering the temperature in Gumbel-Softmax) leads to increased robustness to overfitting, to be less clearly explained, especially how it relates to the first result. It seems obvious to me that, if you can’t pass enough information from the speaker to the listener (when there’s a low temperature), then you won’t be able to solve the task at training time if a lot of information needs to be conveyed (as in the case of randomly shuffled labels). The authors describe this by saying: “With a low temperature (more closely approximating a discrete channel), this is hard, due to stronger entropy minimization pressure.” But this seems misleading to me, since it’s very different from the ‘entropy minimization pressure’ that was discovered in the first result (which comes about when both agents are able to solve the task, but the listener has redundant information). Thus, I don’t see how this result is either surprising or connected to the first result.

Further, the main claim to novelty of this second result is that it tests the information bottleneck principle in a ‘language learning’ set-up. However, since now the communication is continuous, this setup resembles the classic ‘neural network prediction’ even more closely, and calling it a language learning setup seems down to semantics. Given this, I’m unsatisfied with the comparison to previous work on the information bottleneck (which is not my area of expertise).

Given the above points, I’d say the paper is borderline it its current form, with a tendency towards rejection. However, I’d be willing to increase my score if the authors can clarify some of my confusion around the second experiment.

UPDATE: I've increased my review to a 6 after the author rebuttal, although I still feel the paper is borderline.


**Experience Assessment:**

I have published in this field for several years.

**Review Assessment: Checking Correctness Of Derivations And Theory:**

N/A

**Review Assessment: Checking Correctness Of Experiments:**

I assessed the sensibility of the experiments.

**Review Assessment: Thoroughness In Paper Reading:**

I read the paper at least twice and used my best judgement in assessing the paper.

---

> ### Author Response · Authors · 2019-11-07
> **Reply to AnonReviewer3**
>
> Thank you for you review.
>
> a) "I do, however, have a slight concern about the novelty of this result. Given that the environments are very simple (consisting of a single message sent by a speaker and a prediction made by a listener), the line between ‘multi-agent emergent communication’ and ‘neural network with discrete representations’ is very blurred. This is addressed only briefly in the related work section, with a handful of post-2016 papers cited. I’d personally want to see a more expanded related work section that goes into some more depth into some of these papers, to be able to better judge the novelty of this contribution."
>
> We present two fundamental novelties compared to earlier work in the information-bottleneck and discrete representation learning areas. First, we do not *impose* a bottleneck, but we see it naturally emerging when forcing the intermediate representations to be language-like (that is, discrete). We are not engineering entropy minimization, but finding that it independently emerges when representations are discrete. We are not aware of other work that has reported this result. Second, we make a connection between this entropy minimization tendency in our neural networks, and studies in linguistics and cognitive science that have reported a similar entropy minimization constraint in natural language. Establishing this hitherto unnoticed connection might be our most impactful contribution, as it suggests that widely discrete communication systems possess similar information-theoretic properties, that might explain some universal trends in human language. We will try to clarify this point further in the text.
>
> Finally, while in the main text we concentrated on the most basic one-symbol communication scenario, the entropy-minimization effect is also observed when the agents communicate with variable-length messages produced and consumed by GRU or Transformer cells (see Appendix C3). This sequential setup is very specific to language, clearly distinguished from those studied in generic sparse-representation work, and we will highlight this point in the main text.
>
> b)  "I found the second result, that making the representations more discrete (by lowering the temperature in Gumbel-Softmax) leads to increased robustness to overfitting, to be less clearly explained, especially how it relates to the first result. It seems obvious to me that, if you can’t pass enough information from the speaker to the listener (when there’s a low temperature), then you won’t be able to solve the task at training time if a lot of information needs to be conveyed (as in the case of randomly shuffled labels). The authors describe this by saying: “With a low temperature (more closely approximating a discrete channel), this is hard, due to stronger entropy minimization pressure.” But this seems misleading to me, since it’s very different from the ‘entropy minimization pressure’ that was discovered in the first result (which comes about when both agents are able to solve the task, but the listener has redundant information). Thus, I don’t see how this result is either surprising or connected to the first result. "
>
> Thank you for raising this question; indeed, we need to highlight the connection more clearly, and we will do so in revising the paper. In S4, we observe that, once trained, the agents' emergent protocol is subject to entropy minimization pressure. We hypothesize that (1) this pressure is caused by the informational bottleneck that is due to discreteness of the communication system, (2) it might have beneficial regularization properties. In S5, we proceed to verify these two hypotheses. Indeed, we demonstrate that passing large amounts of information becomes harder when the discreteness of the system is increased (addressing (1)) and that it leads to useful regularization properties wrt overfitting and adversarial attacks (addressing (2)).

---

> > ### Comment · AnonReviewer3 · 2019-11-11
> > **Response to authors**
> >
> > Thank you for your response. Indeed, the second set of experiments in Section 4.2 make more sense to me, and I'm willing to update my review to a 6, although I still feel the paper is borderline.

---

### Official Review · AnonReviewer4 · 2019-10-29
**Official Blind Review #4**

**Rating:** 1

**Review:**

This paper sets up a couple discrete communication games in which agents must communicate a discrete message in order to solve a classification task.  The paper claims that networks trained to perform this case show a tendency to use as little information as necessary to solve the task.

I vote to reject this paper.

The experiments are not very interesting and I don't at all agree with the assertion of the paper.  The paper claims the networks use only the entropy necessary to solve the task, but there are two main problems with this assertion.  (1) their own experiments don't support this all that strongly, as in the limit of few hidden bits (left half of the x axis in Figure 1), the networks all had noticeable excess information, and (2) and perhaps most damning the paper applies entropy regularization on the sender during training?  Could it perhaps instead be the fact that the entropy of the sender was penalized as an explicit regularization term that the entropy of the senders messages tended to be small?

I also find the experimental design puzzling.  Why both reinforce and the 'stochastic computation graph' approach?  Treating the receiver's output as binary and stochastic without using the log loss of the bernoulli observation model is just giving up on a good gradient as far as the receiver is concerned.

The experiments done are much to simple and the protocol flawed.

The second set of experiments in Figure 3 were not left to converge, so I'm not sure how we can derive a great deal of insight.  Additionally, relaxing the gumbel softmax channel to being continuous rather than discrete technically ruins any argument that there is an entropy bottleneck present anymore, as theoretically even a single dimensional continuous signal could store an arbitrary amount of information.  If the paper wanted to, it could have upper bounded the mutual information between the input and the message using a variational upper bound.

--------------- Response to Response ---------------------------------

I'm editing here in light of continuing to look at the paper and the responses from the author below.

I have to still argue for a rejection of this paper.

I thank the authors for addressing my comments and I admit that at first I thought the paper was minimizing the entropy during training which would have been particularly bad.  While I was mistaken on that point, I still believe the paper is deeply flawed.

In particular, the paper makes a very bold claim, namely that "We find that, under common
training procedures, the emergent languages are subject to an entropy minimization pressure that has also been detected in human language, whereby the mutual
information between the communicating agent’s inputs and the messages is minimized, within the range afforded by the need for successful communication."  But if we are being honest here, the experiments are very lacking to support such a bold claim.

In particular there was one thing I was worried about upon reading the paper again, and is similar to the point raised by the other reviewers.  In Figure 1, we are shown the entropy only of those networks that have succeeded.  Naturally to succeed, the entropy i the message must be large enough to accomodate the size of the remaining bits we are trying to reconstruct.  That is why Figure 1 includes the dotted line, since the networks must be above that line to have good performance.  And the main evidence for the main claim of the paper is that the trained networks are above that line and arguably close to it.

Now, we know that there are clearly solutions to these tasks (in particular the Guess Number task) which could achieve good performance at noticeably higher entropy.  For instance we could take any minimal solution and simply split up each message into 8x different buckets, each of which had exactly the same behavior from the decoder.  This would give us a +3 in the entropy of our message space while having no effect whatsoever on the loss.  The claim of the paper is that under normal training procedures it seems like we don't find those solutions and instead seem to find minimal ones.

But after implementing a simplified version of the experiment in the paper (Notebook available here: https://nbviewer.jupyter.org/urls/pastebin.com/raw/ZF7g34GN ) I suspect something much simpler is going on.  The reason the solutions look minimal in Figure 1 is probably because the initialization chosen for the encoder they used in the paper tended to start at low entropies.  Imagine if all of the networks started out with an initial message entropy of around 3 bits.  Then Figure 1 could be explained by the problems with hidden bits ~< 3 bits simply preserved their entropy, which in order to solve the task with higher numbers of digits hidden we know requires some minimal budget, so they get sort of pushed up.  This could explain the figure, but we wouldn't claim this explains why we observe small entropy for the high number of hidden digits case.

In particular, if we initialized the encoders with higher entropy, we might expect that we fail to see this phenomenon.  That is exactly what I was able to show for myself in that notebook.  If you simply initialize the encoder to have high entropy, all of the solutions have high entropy and the observed effect goes away.

Overall, the paper as I said is low quality.  Several choices were made that don't make a lot of sense.  With the experiments being as small scale as they were, why not explicitly marginalize out the message (as I did in the notebook)?  Why use single layer neural networks to predict 256 x 1024 parameters? Why not just learn them directly?   If the paper aimed to mimic more standard setups and show that under those setups we observe this kind of minimal message entropy, then it would have to much better tease out the effects of all of these choices.

Why does the decoder use a mean field sigmoid bernoulli observation model to try to predict something is in one of ~32 states?  The missing digits are not independent given the message, why model them as so?  Is that part of the purported reason why these models show minimal entropy (cause it isn't discussed).

For such a simple problem, you could presumably analytically compute the gradient with respect to the loss and study whether that correlates with the gradient of the entropy.  There are several things I could imagine checking, none of which are checked in the paper.

The primary question the paper addresses is an interesting one.  But this paper does very little to carefully investigate that question.  I maintain my vote to reject.

**Experience Assessment:**

I have published in this field for several years.

**Review Assessment: Checking Correctness Of Derivations And Theory:**

N/A

**Review Assessment: Checking Correctness Of Experiments:**

I carefully checked the experiments.

**Review Assessment: Thoroughness In Paper Reading:**

I read the paper thoroughly.

---

> ### Author Response · Authors · 2019-11-06
> **Answer to AnonReviewer4**
>
> Thank you for your review.
>
> a)  "The paper claims the networks use only the entropy necessary to solve the task, but there are two main problems with this assertion.  (1) their own experiments don't support this all that strongly, as in the limit of few hidden bits (left half of the x axis in Figure 1), the networks all had noticeable excess information, and (2) and perhaps most damning the paper applies entropy regularization on the sender during training?"
>
> Answer to (2): Lack of clarity in our technical exposition has led to a fundamental misunderstanding here. The goal of the entropy regularization, as it is often used in Reinforcement Learning [1], is to favor exploration during training. Thus, it is actually applying pressure TOWARDS INCREASING ENTROPY, not minimizing it. As we discuss in S4.1, even in the presence of such an entropy-maximizing term in the loss, the entropy-minimization effect takes place. Moreover, we report a similar behavior in the case of Gumbel Softmax-based training, where no entropy-related term is present in the loss.
>
> We believe that this confusion led AnonReviewer4 to misinterpret our results, and we hope that this clarification (that we will also add to the paper, in section 3.2) will help them to appreciate the non-trivial result we are reporting. Entropy minimization consistently arises without any term in the cost function encouraging it, and, as a matter of fact, with a term that encourages entropy maximization.
>
> Answer to (1): Concerning the interpretation of Figure 1, as we discuss in the text and demonstrate in Appendix, when no bits are hidden, there is no communication, and thus no meaningful analysis of the "communication" protocol. We agree that the bound is less tight as we approximate this limit, possibly because the receiver can rely less on the protocol (and we are thus approaching the no-communication limit). However, here, as in the next experiment (Fig. 2) and all the ablations reported in the main text and appendix, entropy is always very significantly closer to the lower than upper bounds (8 bits in Fig. 1), so we believe our claim fully holds.
>
> b) "Why both reinforce and the 'stochastic computation graph' approach?  Treating the receiver's output as binary and stochastic without using the log loss of the bernoulli observation model is just giving up on a good gradient as far as the receiver is concerned."
>
> Our goal was to cover all major approaches used in the language emergence literature and make sure that our reported entropy minimization effect holds for all of them. Hence, we experimented with Gumbel Softmax relaxation, the Stochastic Computation Graph approach, and REINFORCE. Indeed, the latter does not use a gradient backpropagated from the loss for Receiver, but we believe this only highlights how general our result is.
>
>
> c) "The second set of experiments in Figure 3 were not left to converge, so I'm not sure how we can derive a great deal of insight. "
> Indeed, in the current form of Figure 3, we did not extend the horizontal axis until convergence of all curves. However, we treat speed of learning as a proxy for how easy it is to memorize random labels and, consequently, how tight the informational bottleneck is. We support this connection by running the experiments reported in Figures 3(e) & 3(f).
>
> d) "Additionally, relaxing the gumbel softmax channel to being continuous rather than discrete technically ruins any argument that there is an entropy bottleneck present anymore, as theoretically even a single dimensional continuous signal could store an arbitrary amount of information."
> At training-time, for Gumbel Softmax-based architectures, we apply discretisation noise (standard for Gumbel Softmax training) which prevents the architecture from storing an arbitrary amount of information in the outputs of Sender. However, we believe that AnonReviewer4's observation is close to the argument that we are building in S4.3: as the temperature decreases, the channel becomes increasingly discrete, acting as a tighter bottleneck.
>
>
> [1] Asynchronous methods for deep reinforcement learning, Mnih et al., JMLR 2016

---

### Official Review · AnonReviewer2 · 2019-10-31
**Official Blind Review #2**

**Rating:** 6

**Review:**

What is the specific question/problem tackled by the paper?

The paper studies whether discrete communication channels between agents are low-entropy. The claim is that agents that try to solve a prediction task subject to a communication bottleneck will exchange low entropy messages, even if these messages are not explicitly encouraged to have low entropy.

Is the approach well motivated, including being well-placed in the literature?

The paper is well motivated, though I am not an expert in the area.

Does the paper support the claims? This includes determining if results, whether theoretical or empirical, are correct and if they are scientifically rigorous.

The support for the claims is almost adequate. The entropy of the messages was only analyzed for the runs where agents have successfully learned to communicate in order to solve the task. Part of the paper's conclusion is that an entropy constraint on messages is not necessary, but maybe it still is necessary to increase the frequency of successful runs, or help faster learning.

This means that successful runs lead to low-entropy messages, but what about the unsuccessful runs? Do messages have low entropy as well?

I am also somewhat confused by the second set of experiments. The discussion seems to suggest that setting higher temperature in GS creates pressure for lower-entropy messages. Buf if that's the case, then there's a controllable parameter that implicitly controls an entropy constraint and it's no longer clear to me that low-entropy is emerging.

Summarize what the paper claims to do/contribute. Be positive and generous.

I think the paper does an interesting analysis and makes an interesting point about the problem being studied. I am a bit confused by how the experimental setup supports the claims and their consequences. In particular, I have some doubts about the claim that entropy regularization is unnecessary.

Clearly state your decision (accept or reject) with one or two key reasons for this choice.

I am voting for acceptance.

Provide supporting arguments for the reasons for the decision.

The paper sets up a clear problem to study and focuses on increasing our understanding around the issue. After reading the paper a few times I am a bit confused about how the experimental setup supports the claims & conclusions. I think the results in Figs. 1-2 adequately support the claim, but the results in Fig. 3 make it unclear whether the temperature parameter is implicitly controlling entropy. The fact that unsuccessful runs were discarded for the first set of experiments limits the implications of the main claim that low entropy emerges, because an entropy regularization might still meaningfully improve the frequency of successful runs.

Provide additional feedback with the aim to improve the paper. Make it clear that these points are here to help, and not necessarily part of your decision assessment.

I think if the paper will be improved if it resolves the lack of clarity around the temperature in GS being an implicit entropy-regularization parameter. Perhaps an entropy-regularized setup is a useful comparison to show that it provides marginal benefit over the setup studied, and this might resolve the lack of clarity around the implications of the claims made from the first set of experiments.

Apart from these issues I am happy with the choice of topic and execution of the paper. I also appreciate that due care has been taken to present the work as understanding a phenomenon, to avoid any misconceptions about a new method being proposed.

**Experience Assessment:**

I do not know much about this area.

**Review Assessment: Checking Correctness Of Derivations And Theory:**

N/A

**Review Assessment: Checking Correctness Of Experiments:**

I assessed the sensibility of the experiments.

**Review Assessment: Thoroughness In Paper Reading:**

I made a quick assessment of this paper.

---

> ### Author Response · Authors · 2019-11-07
> **Answer to AnonReviewer2**
>
> Thank you for your feedback! Please find our comments below.
>
> a) “Part of the paper's conclusion is that an entropy constraint on messages is not necessary, but maybe it still is necessary to increase the frequency of successful runs, or help faster learning.”
>
> Following standard practice, we do add an entropy *maximization* term to our cost function, to encourage exploration during learning (except when training with Gumbel Softmax). However, even when this term is included, the emergent protocol tends to have *low* entropy (see our response to AnonReviewer4). This term is definitely important to increase the frequency of successful runs. We would not advocate removing it, and it is unrelated to the entropy *minimization* property that we observe naturally emerging in all our simulations.
>
> Our claims are that 1) entropy minimization is an emergent property of discrete communication between neural networks, observed across a variety of experimental setups; 2) since the same pattern has been observed in natural language, we establish an important parallelism in the information-theoretic properties of different discrete communication systems. Our experiments on robustness offer initial evidence that this property might have independent beneficial properties, which might even provide some insight on why human language evolved to be discrete. We will clarify our line of reasoning in the conclusion, as we see that it is too implicit in the current version.
>
>
> b) “I am also somewhat confused by the second set of experiments. The discussion seems to suggest that setting higher temperature in GS creates pressure for lower-entropy messages. Buf if that's the case, then there's a controllable parameter that implicitly controls an entropy constraint and it's no longer clear to me that low-entropy is emerging.”
>
> In the GS setup, temperature controls how tight the approximation to a discrete channel is, not entropy per se. In the relevant experiments, we show that the degree of entropy minimization closely follows the tightness of the discrete approximation. We conclude that emergent entropy minimization in the fully-discrete setup studied in the previous experiments must indeed be connected to the fact that the channel is discrete. Again, this caused some confusion to other reviewers as well, and we will outline the reasoning behind the experiments more clearly in the paper.
>
> c) “I think the results in Figs. 1-2 adequately support the claim, but the results in Fig. 3 make it unclear whether the temperature parameter is implicitly controlling entropy. The fact that unsuccessful runs were discarded for the first set of experiments limits the implications of the main claim that low entropy emerges, because an entropy regularization might still meaningfully improve the frequency of successful runs.”
>
> Concerning the relation between temperature and entropy, please see above.
>
> Can you kindly clarify your next point? Note that we do use entropy regularization where appropriate, but this is actually a term that enforces entropy *maximization*, making the emergence of entropy minimization even more intriguing (again, please see our response to AnonReviewer4).

---

> > ### Comment · AnonReviewer2 · 2019-11-09
> > **Still some concerns over unsuccessful runs, and the connection of the second set of experiments with the rest of the paper**
> >
> > Looking at Fig 1 (c) and (d), I see that the entropy of the message increases as the weight of the entropy bonus increases, except for the largest values in each plot, which have entropy closest for the smallest (!). Is this perhaps because of discarding unsuccessful runs?
> >
> > (a) Thanks for the clarification. Looking at Fig 1 (c) and (d), I see that the entropy of the message increases as the weight of the entropy bonus increases, except for the largest values in each plot, which have entropy closest for the smallest. Is this perhaps because of discarding unsuccessful runs? Apart from that, it does seem that REINFORCE responds to regularization by increasing entropy, but not by much. Perhaps the conclusion is that performance is driving the entropy down, in spite of the entropy regularization. This is in line with the claim (1) (in the rebuttal).
> >
> > I still think that considering only successful runs and discarding unsuccessful ones is an issue, and it makes curves hard to adequately compare. I discuss some ways to try to mitigate this below (in c).
> >
> > (b) The results support the claim that increasing the GS temperature acts as a regularizer. With less regularization, the effective set of hypotheses is richer, which means more overfitting. (I also think the magnitude of scores must be implicitly controlled by the training and use of neural networks, otherwise one could make up with large weights for restrinctions imposed by larger temperatures).
> >
> > Could you please clarify the beginning of section 4.2 to provide a clearer statement of the main claim for the second set of experiments? The hypothesis seems to be that "information bottlenecks are important/relevant/partially responsible for emergence of low-entropy messages in human communication." Then you proceed to show this in the artifical settings being considered, since evidence in the artifical setting can be indirect evidence to the claim about the natural setting. But the outcome of the experiments seem to align with the claim that "one can communicate less information as the information bottleneck becomes stronger."
> >
> > I share the impression of AnonReviewer3 that the results in the second set of experiments are not clearly connected to the ones in the first set of experiments, so this connection should be presented more clearly. That is, how are the results connected to the claims about emergence of low-entropy communication?
> >
> > (c) I had partially misunderstood the experimental setup, in particular that there was an entropy bonus in place. My concern is that the behavior in failed runs is not being considered. It would not be too much of an issue if the entropy in failed runs were high, in which case you could still argue that performance drives low-entropy. Also in this case I would expect the failed runs to increase (at least in REINFORCE) with the weight of the entropy bonus. The problem is if the messages are low-entropy also in the case of failure.
> >
> > I have some other thoughts about mitigating the discarded run issues. You could consider looking at the point plots for entropy versus performance. Maybe the pressure for minimizing entropy is present in all but the lowest performance levels. In that case you should also see the setups with higher entropy bonus have lower performance than the ones with less entropy. There's exploration related issues that can affect the observations. If some amount of exploration is needed to solve the task, then "too little" entropy bonus can be associated with bad performance too.

---

> > > ### Author Response · Authors · 2019-11-11
> > > **Answer: Still some concerns over unsuccessful runs, and the connection of the second set of experiments with the rest of the paper**
> > >
> > > Thank you for the discussion and the suggestion! This is very interesting.
> > >
> > > (a) "It would not be too much of an issue if the entropy in failed runs were high, in which case you could still argue that performance drives low-entropy. Also in this case I would expect the failed runs to increase (at least in REINFORCE) with the weight of the entropy bonus. The problem is if the messages are low-entropy also in the case of failure."
> > >
> > > As far as we understand, your concern is the following scenario. At first, before any training, all models start with a high-entropy communication protocol. Then, during training, (potentially) some models would increase entropy of their messages, and (potentially) some would decrease. In a "problematic" scenario, all runs, either successful or not,  would have their entropy decreased, possibly indicating that entropy minimization has nothing to do with model's training success.
> > >
> > > We think that it’s unlikely that the entropy level of the failed runs is diagnostic of a problem: Even unsuccessful models are optimized for communication success and they might achieve a partial solution and, hence, it would be unclear how the low entropy of failed runs could be interpreted. The training process seems to be a complex interplay between agents' mutual coordination, exploration, and optimization (see below), rather than filtering the runs that have sufficiently high or low entropy.
> > >
> > > Further, note that while REINFORCE and Stochastic Computational Graph training involves a bonus for entropy, it is typically small and dominated by the communication reward. With a higher bonus, the runs do not achieve communication success, due to the entropy term dominating the loss.
> > >
> > > Inspired by your suggestion, we performed a series of experiments, reported in Appendix F of the updated draft. In these experiments, we report how entropy of the messages and accuracy changes during training for a few sampled successful and failed runs.
> > >
> > > The obtained results indicate that the training story is rather complicated. Firstly, Sender's initial message entropy can be higher (Figures 10(a) and 11(a)) or lower (Figures 10(d) and 11(d))
> > > than required for communication success. In the latter case, the resulting protocol's entropy is higher than at the start, for both successful and unsuccessful runs. In the former case, the resulting entropy can fall w.r.t. the initial one. Both successful and unsuccessful trajectories can display entropy climbs and falls; sometimes successful runs have entropy below, sometimes - above the unsuccessful runs.  There is no clear relation between entropy level and success (apart from the fact that insufficient entropy would correlate with a failure).
> > >
> > > Our interpretation of the observed patterns is that agents have a hard time discovering a high-entropy protocol, and/or coordinating successful communication through it. Consequently, if they hit a high-entropy spot, they drift towards a lower-entropy one, and only there they can converge on successful runs. These dynamics lead to the observed bottleneck.
> > >
> > > (b) "I share the impression of AnonReviewer3 that the results in the second set of experiments are not clearly connected to the ones in the first set of experiments"
> > >
> > > We have updated Section 4.2 to provide a better connection between the two experimental parts of the paper. We also added various improvements, addressing other comments.

---

### Author Response · Authors · 2019-11-11
**Revision submitted**

We have updated the text, addressing various comments. Most importantly, we (1) clarified the connection between Sections 4.1 and 4.2, (2) clarified the role of the entropy regularization (maximization) term, (3) added a comment on the role of the Receiver-side entropy regularization weight $\lambda_r$ in REINFORCE, (4) highlighted the difference with the representation learning literature.

---

### Decision · Program_Chairs · 2019-12-19

**Decision:**

Reject

**Comment:**

This paper studies the information-theoretic complexity for emergent languages when pairs of neural networks are trained to solve a two player communication game. One of the primary claims of the paper was that under common training protocols, networks were biased towards low entropy solutions. During the discussion period, one reviewer shared an ipython notebook investigating the experiments shown in Figure 1. There it was discovered that low entropy solutions were only obtained for networks which were themselves initialized at low entropy configurations. When networks are initialized at high entropy configurations, the converged solution would remain high entropy. This experiment raises questions about the validity of the claim that there was "pressure" towards low entropy solutions to the task. Therefore, a more careful analysis of the phenomenon is required.